# Physical-Mechanical Properties of Cupola Slag Cement Paste

Carlos Thomas *, José Sainz-Aja, Israel Sosa, Jesús Setién, Juan A. Polanco and Ana Cimentada

LADICIM (Laboratory of Materials Science and Engineering), University of Cantabria, E.T.S. de Ingenieros de Caminos, Canales y Puertos, Av. Los Castros 44, 39005 Santander, Spain; jose.sainz-aja@unican.es (J.S.-A.); israel.sosa@unican.es (I.S.); setienj@unican.es (J.S.); polancoa@unican.es (J.A.P.); cimentadai@unican.es (A.C.)
* Correspondence: thomasc@unican.es; Tel.: +34-942-200-915

**Abstract:** The high consumption of natural resources in the industrial sector makes it necessary to implement measures that enable the reuse of the waste generated, seeking to achieve circular economy. This work assesses the viability of an alternative to the use of CEM III B 32.5 R cement in mortars for the internal coating of centrifugally spun cast iron pipes for water piping. The proposal is to reuse the slag generated in the casting process after being finely ground, as an addition mixed with CEM I 52.5 R cement, which is basically Portland clinker. In order to analyse this possibility, an extensive experimental campaign was carried out, including the analysis of the cupola slag (micro-structural and chemical composition, leachates, setting time, vitrification, puzzolanicity and resistance to sulphate) and regarding the mortars (workability and mechanical properties). The experimental programme has shown that the optimum substitution is achieved with a replacement percentage of 20% of the cement, with which similar workability, superior mechanical properties and guaranteed resistance to sulphate attack are obtained. In addition, both economic and environmental savings are achieved by not having to transport or landfill the waste. In addition, the new cement is cheaper than the cement currently used.

**Keywords:** cupola slag; mortar; sustainability; waste recovery; circular economy

## 1. Introduction

The changes introduced in the iron and steel manufacturing processes of ferrous alloys have generated, in recent years, new types of industrial waste which, given the unavoidable requirements of sustainability, must be reused as much as possible. In particular, steel slag has recently been the subject of numerous valorisation studies [1–3] that have accredited its suitability for different uses depending on its origin and characteristics; these range from being used as a raw material in the production of cement to its use as aggregates for concrete or bituminous mixtures [4,5].

Slags can be produced from a wide range of metallurgical sources, such as iron, steel, nickel, manganese, copper, etc. The various types of slags have similar metallurgical functions, although they vary widely in their physical and chemical properties. The most commonly used slag in the construction industry is steel slag, which comes from steelmaking and smelting. Traditional steel production takes place in blast furnaces and the raw materials used are iron ore, coke and lime, limestone or dolomite as fluxes. The slag thus generated is called blast furnace slag, defined as "a non-metallic product consisting essentially of silicates and aluminosilicates of calcium and other bases, which was developed under liquid conditions together with iron in the blast furnace". Currently, other options have been established based on three types of processes: electric arc furnace, basic oxygen furnace and ladle furnace. The raw materials for steelmaking are iron and blast furnace slag, in varying proportions depending on the furnace, lime and limestone or dolomite.

Blast furnace slag has been used in the binder industry for a long time, and it is interesting to note that as early as 1862 the first known tests on the granulation of slag were carried out, showing that this basic slag, ground and mixed with hydraulic lime, gave

rise to a material which, without reaching the quality of Portland cement, was superior to systems which only used lime as a binder. This waste is also used in mining technologies such as backfilling, which is desirable for environmental reasons [6].

In the case of electric arc furnace slag, the existence of certain previous experiences related to its reuse as a construction material, carried out without prior control of the quality of the waste, and which in some cases have posed serious problems for the safety of the construction site and/or the environment, highlight the need to carry out a specific and detailed investigation to guarantee the suitability of the proposed application.

Cupola slag (CS) is an industrial by-product generated in the production of parts after a casting process in cupola furnaces. In general, for each tonne of molten metal, between 50–60 kg of CS are generated [7], which, in Spain, is mostly sent to landfill. It is a waste that is easily separated from the metal by flotation when both are in a liquid state due to the considerably lower density of the slag. The properties of these slags will depend, to a large extent, on the cooling process, since the faster the cooling process, the more chemical compounds will present an amorphous structure, i.e., the more susceptible the slag will be to pozzolanic, active behaviour.

Concrete is the most widely used material in construction and is estimated to be responsible for 10% of man-made $CO_2$ emissions [8–10]. For this reason, many solutions have been proposed over the years to reduce the environmental impact of concrete manufacture [11–13]. One of these options is to recover waste by using it as aggregates for the manufacture of concrete. Although the most commonly used waste is construction and demolition waste [14–17], there are also a large number of studies in which industrial by-products, such as slag, are incorporated as aggregates [2,18]. Another of the options proposed to achieve the most environmentally friendly concrete has been to recover waste as fines [17,19,20], even generating recycled concrete cements [21,22].

There are very few studies that consider the possibility of recovering cupola slag in concrete. Baricová et al. [23] analysed the use of cupola slag as aggregates, both fine and coarse, and observed a large reduction in the mechanical properties of concrete. On the other hand, there are several studies on the use of cupola slag as a replacement for cement that show contradictory results. Mistry and Varia [24] also analysed and concluded that cupola slag can be used as coarse aggregate effectively in structural as well as in plain concrete. Alabi and Mahachi [25] concluded that the cupola slag aggregate concrete shows a satisfactory development and consistency in strength as compared to NAC. Pribulova et al. [26] concluded that the use of cupola slag aggregates was possible to manufacture high density concrete while the use of the cupola granulated slag as the replacement of granulated blast-furnace slag in the production of cement-free concrete has not proved to be suitable. Aderibigbe et al. [27] analysed whether cupola slag was reactive, but found that it was a residue with very little pozzolanic activity and limited the percentage of Portland cement replacement (OPC) to 20%, which implied a reduction in concrete properties of 13.5%. Stroup et al. [28] tested the effect of 35% OPC substitution with cupola slag and obtained an increase in compressive strength of 8%. Ceccato et al. [29] analysed different substitution ratios and different water/cement (*w/c*) ratios, concluding that the effect of cupola slag is not seen until advanced ages and that for substitutions up to 10% there is no loss of mechanical properties. This 10% limit was also supported by Afolayan et al. [30]. There are also a number of studies in which cupola slag is used as an admixture, obtaining concretes with good mechanical and durability properties [3,4,31].

For the production of ductile cast iron parts, such as the manufacture of water pipes, an internal mortar coating is also required to guarantee drinkability. In fact, this internal coating prevents any possible oxides generated in the pipes from passing into the water supply systems. In this study, the possibility of valorising the slag generated in the manufacturing process of ductile iron pipes by incorporating it into the mortar used in the inner lining is considered, in a notable exercise of circular economy. The starting hypothesis is to replace the iron and steel cement currently used with another cement made from the combination of an OPC and this finely ground slag. The result must necessarily give rise to

a sulphate-resistant cement, which is a requirement for use in the internal lining of pipes for water distribution networks.

## 2. Materials and Methods

### 2.1. Materials

2.1.1. Cement

In this work, two types of cement were used: CEM I-52.5 R and CEM III-32.5N-SR according to EN 197-1 [32]. The density and Blaine specific surface area were determined according to UNE 80103 [33] and EN 196-6 [34] respectively. In addition, the chemical composition of each cement was determined by means of an X-ray fluorescence test.

2.1.2. Cupola Slag

The chemical composition of the slag is essential to define its cementing properties. In addition, it was considered essential to check that this chemical composition was stable over time, so periodic tests were carried out for 13 months.

Subsequently, in order to assess the use of slag as a replacement for OPC in the manufacture of mortar used in the internal lining of pipelines, it is necessary to ensure that it does not contain chemical compounds that could be harmful to health. To check this point, a leachate test was carried out in order to obtain the concentration of the chemical elements present in the slag. The analysis of the cupola slag leachate was carried out in accordance with EN 12457-4 [35].

Once it was verified that the use of the cupola slag does not suppose any potential risk, cupola slag filler (CSF) was obtained from cupola slag aggregates (CSA). The process of obtaining CSF consists of 7 phases, and images of the complete reduction process can be seen in Figure 1.

- Phase 1 (Figure 1a): CSA obtained from the pipe manufacturing process is available.
- Phase 2 (Figure 1b): The CSA is fed into a ball crusher with water for 8 h.
- Phase 3 (Figure 1c): After 8 h of wet crushing, the contents of the mill are poured into a large container.
- Phase 4 (Figure 1d): To facilitate drying, the material is transferred to smaller trays with a larger specific surface area.
- Stage 5 (Figure 1e): The trays are placed in an oven at 100 °C until the material is found to be free of moisture.
- Phase 6 (Figure 1f): Once the material is found to be completely dry, it is ready for use.
- Phase 7 (Figure 1g): Final appearance of the CSF.

It was decided to crush 8 h after testing different times and obtaining the minimum time required to reduce the material size to filler. 2 h, 4 h, 6 h, and 8 h were tested and it was found that it is necessary 8 h, in an enamel mill with alumina balls, to obtain more than the 95%wt. of the material smaller than 75 μm.

The characterisation of CSF was divided into five phases; physical properties, setting time, vitrification, pozzolanicity test and ensuring sulphate resistance. The physical properties were obtained in the same way as for cement. To ensure that the actual pipe manufacturing process is not significantly modified, the setting times were compared using the procedure described in EN-196-3 [36]. Initially, three mixtures were used: the reference cement CEM III; a mixture of 80% CEM I and 20% CSF; and a mixture of 50% CEM I and 50% CSF. In a second phase, the mixture of 50% CEM I and 50% CSF was also used, but using an activator (1% sodium oxide in relation to the amount of CSF added). To analyse the degree of vitrification of the cupola slag, an X-ray diffractogram was performed. Once the presence of amorphous silica compounds was verified, a pozzolanicity test was performed according to EN-196-5 [37]. Finally, as the application requires resistance to sulphates, and the CEM III currently used to line pipes is sulphate resistant, it was necessary to check that the mixtures of CEM I with CSF are also sulphate resistant. For this purpose, the amount of $C_3A$ in the mixture was calculated to check that the mixture meets the sulphate resistance

requirement of EN 197-1 [32]. For this purpose, the Bogue Formula (1), was used, which was applied to the CEM I clinker, as CSF does not provide C$_3$A.

$$C_3A = -1.6920Fe_2O_3 + 2.6504Al_2O_3 \qquad (1)$$

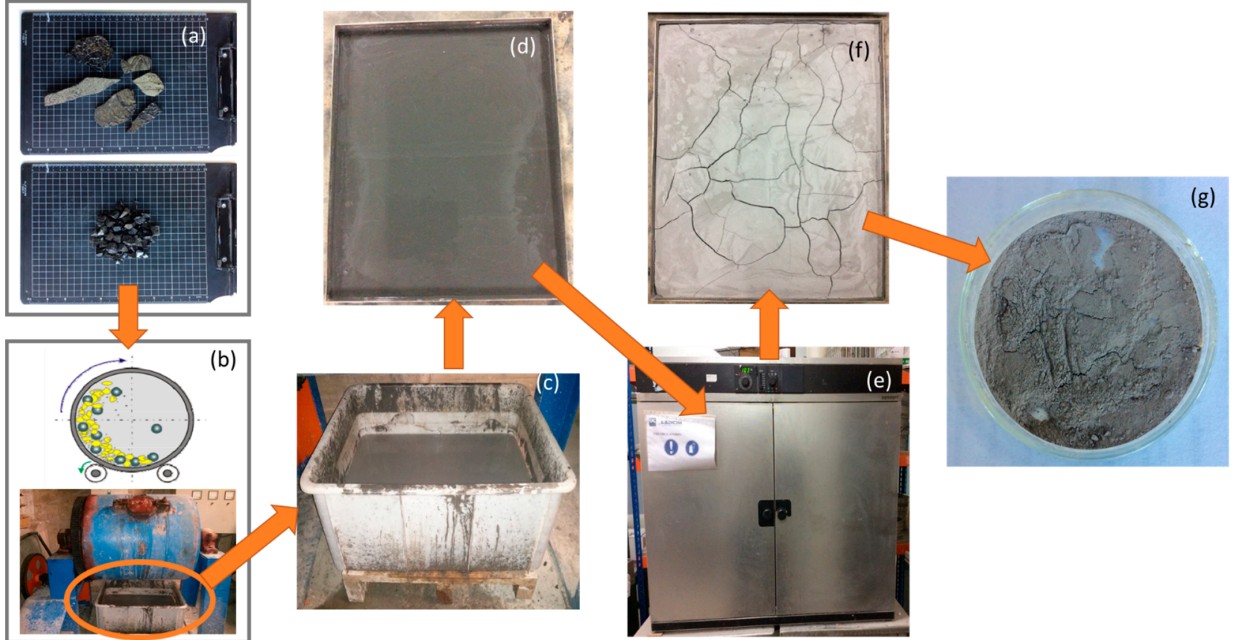

**Figure 1.** Cupola slag manufacture process.

### 2.1.3. Aggregates

The aggregates used in the tests in this study are fine aggregates: silica sands with a density of 2650 kg/m$^3$. The characterisation of the aggregates used for the manufacture of the mortars is carried out by considering the macroscopic characteristics, thus carrying out the geometric and dimensional characterisation; specifically, the grading curve of the sand used was determined, see Figure 2, on the basis of the EN 933-1 standard [38]. The procedure starts by selecting a sample of material, which is introduced into the oven until it reaches a constant weight. Once the sample is completely dry, it is sieved and each of the fractions obtained is weighed.

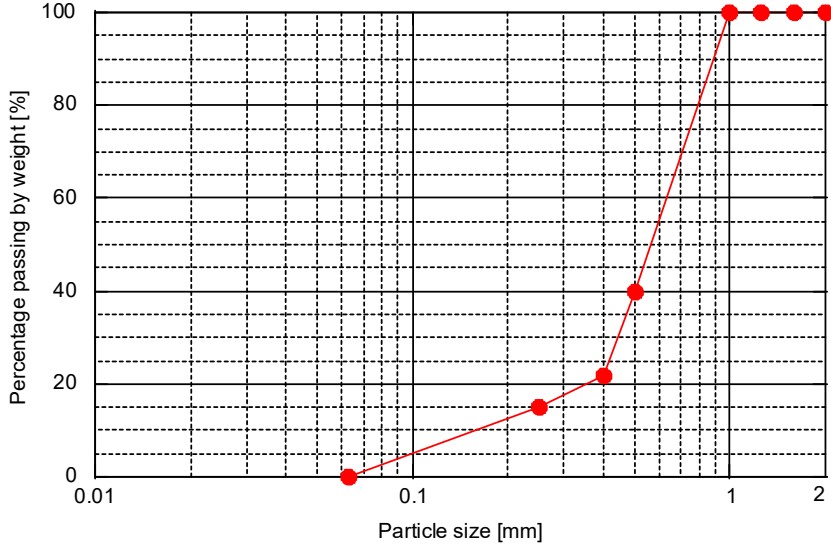

**Figure 2.** Sand grading curve.

### 2.2. Methods

#### 2.2.1. Mix Proportions

In order to analyse the possibility of replacing the commercial cements currently used with a combination of CEM I and CSF, eight different mixes were designed. The first mix (M-Rf) is the commercial mix used to manufacture the mortar to coat the inside of cast iron pipes, which will be used as a reference. Second, to analyse the possibility of using a 20% replacement of cement by CSF, three mixes were carried out with different *w/c* ratio values. Finally, to analyse the possibility of a 50% replacement of CEM I by CSF, four mixes were designed with different *w/c* ratio values. These *w/c* ratios are low in order to guarantee the dry consistency required for the mortars to be placed by centrifugation. Table 1 shows the mix proportions of the different mixes used. In all cases the same volume of mortar was made, this is the reason why as the *w/c* ratio increases the cement quantity decreases.

**Table 1.** Mortar mix proportions.

|  | M-Rf | M-20-A | M-20-B | M-20-C | M-50-A | M-50-B | M-50-C | M-50-D |
|---|---|---|---|---|---|---|---|---|
| CEM III-32.5N [g] | 317.5 | 0 | 0 | 0 | 0 | 0 | 0 | 0 |
| CEM I-52.5R [g] | 0 | 253.2 | 252.4 | 251.6 | 158.2 | 157.7 | 157.2 | 156.7 |
| Cupola slag (CSF) [g] | 0 | 63.3 | 63.1 | 62.9 | 158.2 | 157.7 | 157.2 | 156.7 |
| Sand [g] | 571.4 | 569.6 | 567.8 | 566.0 | 569.6 | 567.8 | 566 | 564.3 |
| Water [g] | 111.1 | 113.9 | 116.7 | 119.5 | 113.9 | 116.7 | 119.5 | 122.3 |
| Fine particles [g] | 317.5 | 316.5 | 315.5 | 314.5 | 316.5 | 315.5 | 314.5 | 313.5 |
| Sand/cement | 1.8 | 1.8 | 1.8 | 1.8 | 1.8 | 1.8 | 1.8 | 1.8 |
| Water/Cement | 0.35 | 0.36 | 0.37 | 0.38 | 0.36 | 0.37 | 0.38 | 0.39 |

In the first phase of this work, fresh state behaviour tests were carried out using the eight dosages previously mentioned. With these values, in order to guarantee that the replacement of the original mortar with the new mortar enhances the CS, the mortar selected was the one with fresh state behaviour as similar as possible to the reference mortar behaviour. It was decided to use this criterion because the behaviour in the fresh state is a critical parameter due to the manufacturing process of the pipes. In a second phase where the mechanical properties of the manufactured mortars are analysed, only three mixes of the reference mix (M-Rf) and the mix with the behaviour in the fresh state most similar to the M-Rf of each of the replacement mixes were used, namely M-20-C and M-50-D, which in this phase will be called M-20 and M-50 respectively.

#### 2.2.2. Mortar Manufacture, Curing Conditions and Testing Procedure

The manufacturing process of the mortars was as defined by EN-196-1 [39], see Figure 3a. Twenty-four hours after the mortars were manufactured, they were demoulded and placed in a humidity chamber for curing, Figure 3b.

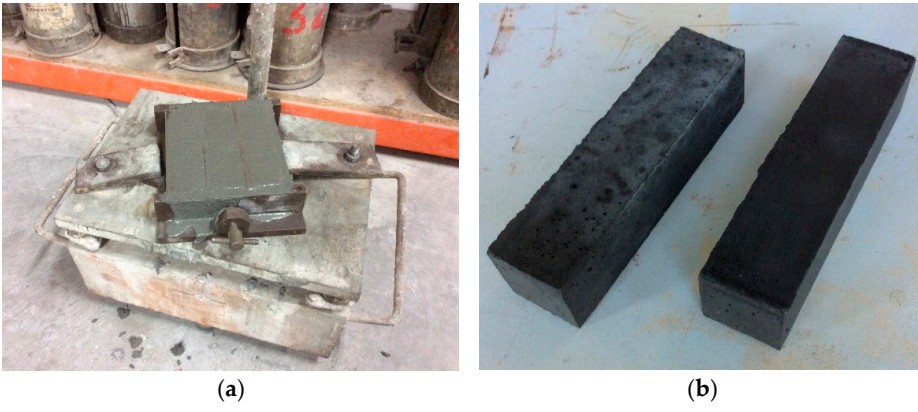

(**a**) (**b**)

**Figure 3.** Mortar manufacture (**a**). Mortar specimens (**b**).

For the determination of the consistency of the mortars, the procedure according to EN 1015-3—Methods of test for mortar for masonry—Part 3: Determination of consistency of fresh mortar (by flow table) [40] is followed. Very similar to that used to determine the consistency of concrete by means of a shaking table. In the case of concrete [41], there is a limitation regarding the maximum size of the aggregate of 64 mm. In the case of mortar, the limitation is 4 mm. In both tests, the fresh cone is shaken 15 times to finally measure the diameter in two perpendicular directions. Figure 4 shows an example of the parameters registered in each test (d1 and d2).

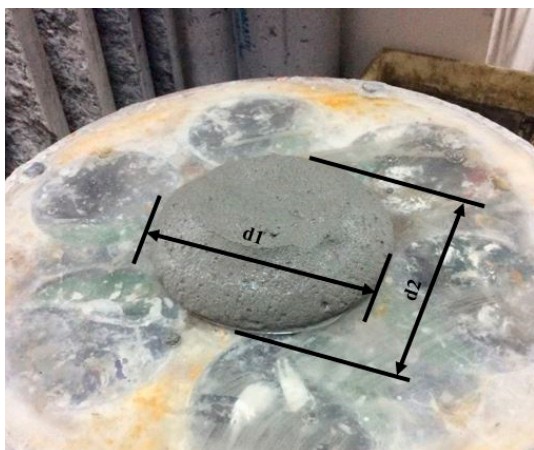

**Figure 4.** Mortar workability test measurement process.

The compressive strength was determined according to EN 196-1 [39] (prismatic specimens of 160 × 40 × 40 mm and at a loading rate of 50 Nm/s). An example of the compressive strength test set-up is shown in Figure 5.

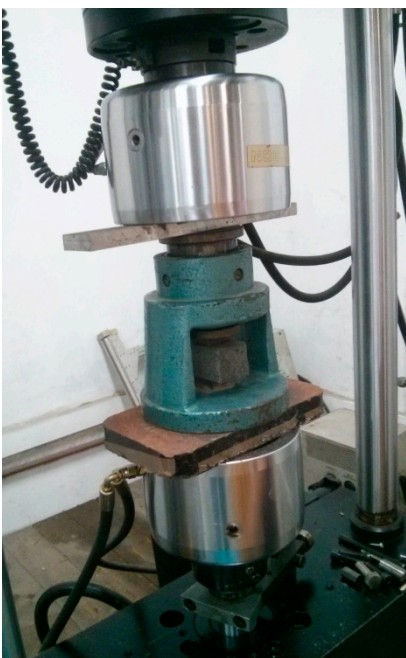

**Figure 5.** Mechanical tests, compressive strength.

## 3. Results

### 3.1. Cement

Table 2 shows the values of the physical properties of the two cements characterised. Table 3 shows the chemical composition of these same cements. The values obtained are as expected for each type of cement.

**Table 2.** Physical properties of the different cements.

| Cement | Density [g/cm$^3$] | Blaine Surface [cm$^2$/g] |
| --- | --- | --- |
| CEM III-32.5N-SR | 2.98 | 3928 |
| CEM I-52.5R | 3.09 | 4166 |

**Table 3.** Chemical composition of the different cements.

| CEM III-32.5N-SR | | CEM I-52.5R | |
| --- | --- | --- | --- |
| Element | Weight (%) | Element | Weight (%) |
| CaO | 50.3 | CaO | 66.6 |
| SiO$_2$ | 29.5 | SiO$_2$ | 17.81 |
| Al$_2$O$_3$ | 9.01 | Al$_2$O$_3$ | 4.79 |
| MgO | 4.76 | SO$_3$ | 4.49 |
| SO$_3$ | 2.29 | Fe$_2$O$_3$ | 3.38 |
| Fe$_2$O$_3$ | 1.21 | MgO | 1.30 |
| K$_2$O | 0.61 | K$_2$O | 0.78 |
| TiO$_2$ | 0.43 | TiO$_2$ | 0.20 |
| Na$_2$O | 0.43 | | |
| MnO | 0.26 | | |

### 3.2. Cupola Slag Characterisation

3.2.1. Microstructural and Chemical Composition

Figure 6 shows the microstructure of the slag in which the maximum particle sizes are checked.

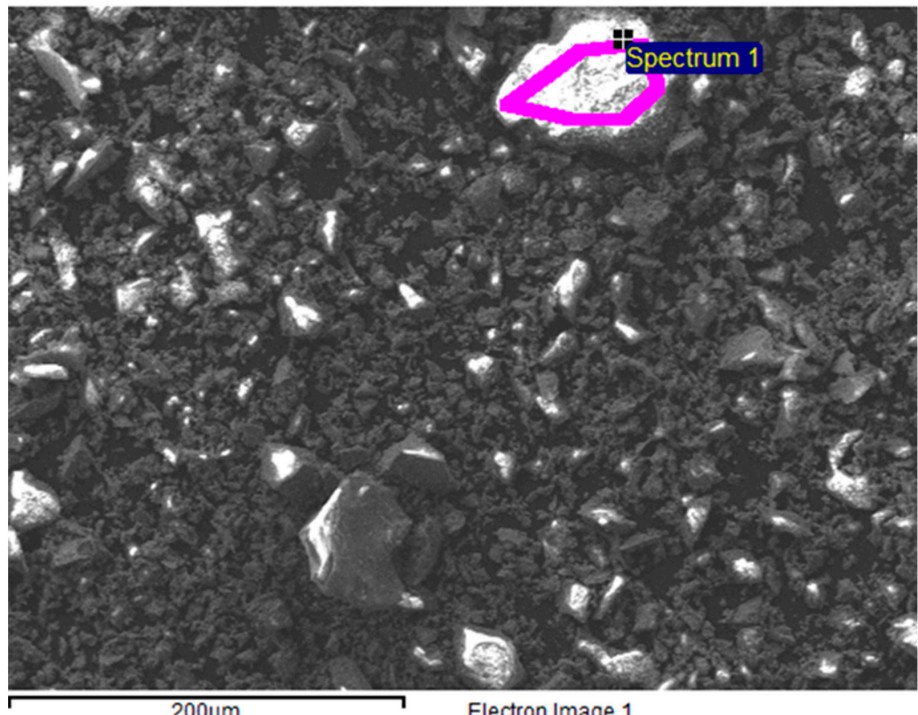

**Figure 6.** Cupola slag microstructure.

An example of the results of the chemical composition analysis performed on cupola slag samples restricted to its elemental components is shown in Figure 7 and Table 4 below.

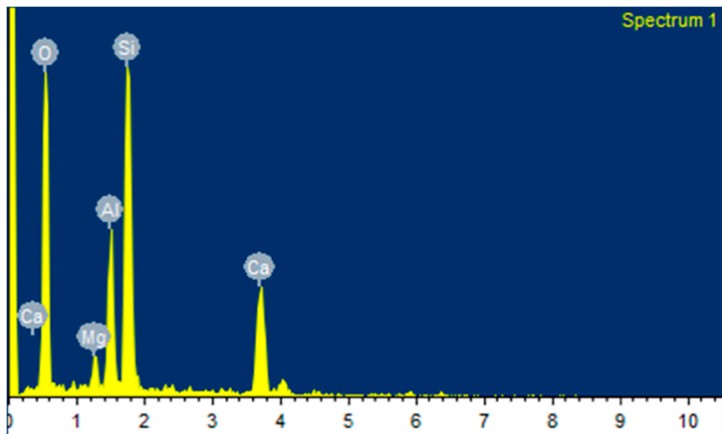

**Figure 7.** Cupula slag chemical analysis.

**Table 4.** Chemical composition of the cupola slag (principal components).

| Element | Weight (%) |
| --- | --- |
| O | 46.59 |
| Mg | 1.56 |
| Al | 7.49 |
| Si | 21.38 |
| Ca | 22.99 |

The casting process is a highly complex process involving a large number of variables. For this reason, slight variations in the chemical composition of cupola slag are expected. Therefore, a study of the evolution of the different components present in the cupola slag over time was carried out. To make this analysis easier, these components were divided into two groups according to the quantity present in each of them: main components and secondary components, see Figure 8. In this figure, the Y-axes indicate the quantity of each component, see legend. For the X-axis, this is a linear scale of the time indicated at the bottom of the figure.

The graphs above show the evolution, over time, of the components present in the cupola slag that can be used to replace cement. The left graphs show the evolution of in time of CaO, $SiO_2$, $Al_2O_3$, MgO and $Fe_2O_3$, while the right graphs show the evolution in time of $K_2O$, $P_2O_5$, MnO, $SO_3$, $Cr_2O_3$ and $TiO_2$.

In the case of the principal components, it can be seen that $Al_2O_3$ remains practically constant, with values between 10 and 15% by weight, as well as MgO and $Fe_2O_3$, whose values over time are around 5% by weight, with some peaks. CaO and $SiO_2$ show somewhat more variability, but within acceptable limits. The greatest variability is observed in the secondary components, but their effect is minor for our purposes. In case of the secondary components, it can be seen that except MnO and specially $SO_3$ the other components are quite stable on time.

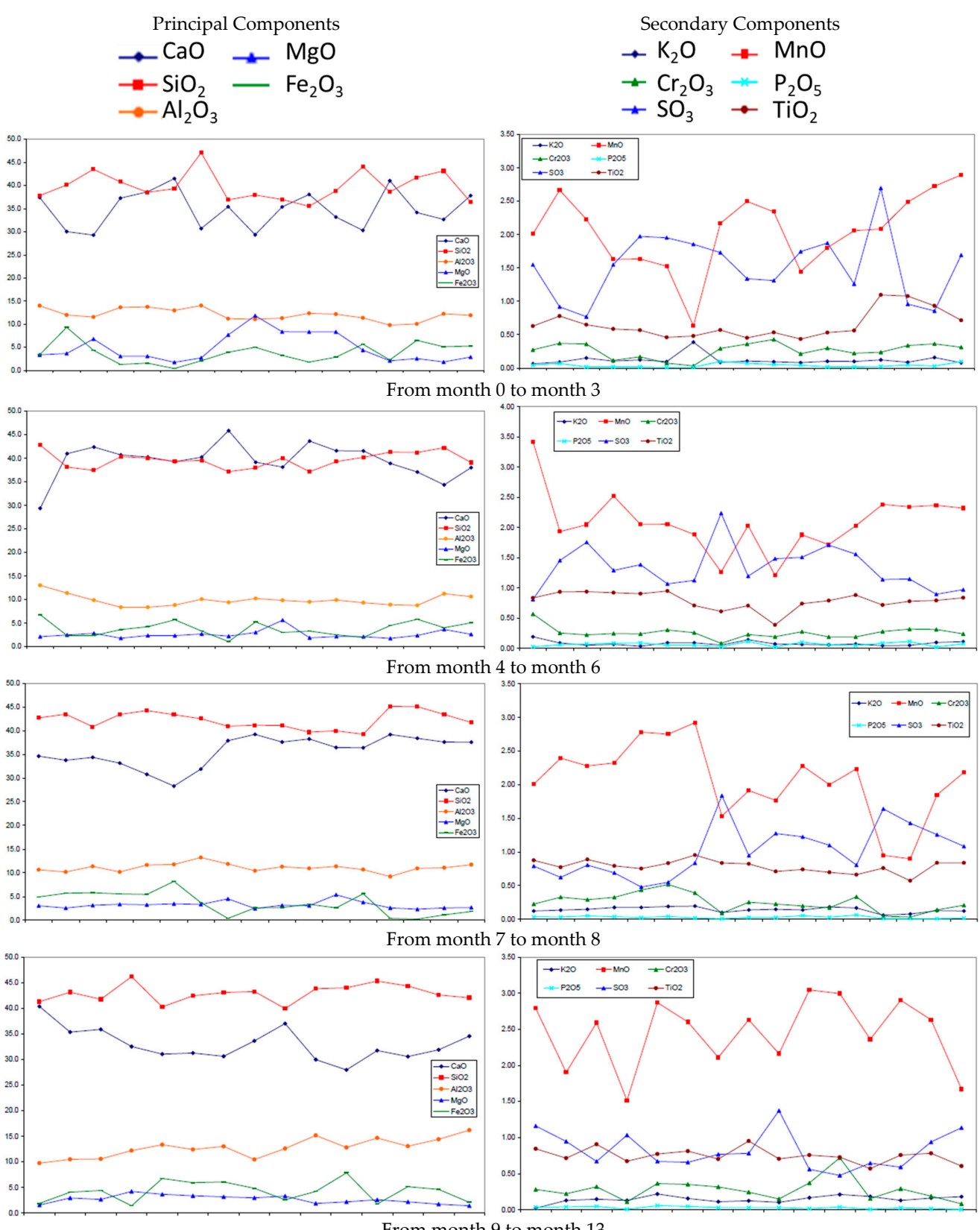

**Figure 8.** Cupola slag chemical composition stability (% wt.).

### 3.2.2. Leachates

The results obtained in the leachate test together with the limit values provided by the local regulations are shown in Table 5.

**Table 5.** Leachate test results.

| Parameter | Unit | Limit Value | Test Results | Parameter | Unit | Limit Value | Test Results |
|---|---|---|---|---|---|---|---|
| Humidity | % | - | 0.1 | Barium | mg/kg m s. | 20 | <0.10 |
| pH | | - | 10.41 (21.6 °C) | Cadmium | mg/kg m s. | 0.04 | <0.01 |
| Conductivity | μS/cm | - | 72.8 (20 °C) | Copper | mg/kg m s. | 2 | <0.17 |
| COD | mg/kg m s. | 500 | <50 | Total chromium | mg/kg m s. | 0.5 | <0.06 |
| Chlorides | mg/kg m s. | 800 | <50 | Mercury | mg/kg m s. | 0.01 | <0.002 |
| Phenols | mg/kg m s. | 1 | <1 | Molybdenum | mg/kg m s. | 0.5 | <0.20 |
| STD | mg/kg m s. | 4 | 480 | Nickel | mg/kg m s. | 0.4 | <0.08 |
| Sulphates | mg/kg m s. | 1000 | 60.4 | Lead | mg/kg m s. | 0.5 | <0.12 |
| Antimony | mg/kg m s. | 0.06 | <0.01 | Zinc | mg/kg m s. | 4 | <0.55 |
| Arsenic | mg/kg m s. | 0.5 | <0.02 | Selenium | mg/kg m s. | 0.1 | <0.10 |

As stated by P. Rodrigues et al. [42,43], the leachates tests is a good method to evaluate the environmental risks. It can be seen that in this case, all the values detected for each chemical element are within the range defined by the aforementioned legislation. For this reason, it is a material that can be used in elements such as water supply networks. Furthermore, as it is intended to be used for the manufacture of mortars, the slag particles will be encapsulated in the cement matrix, so it is presumable that the leachate values of the mortars will be even lower.

### 3.2.3. Physical Properties

Table 6 shows the physical properties of CSF.

**Table 6.** Physical properties of cupola slag.

| Material | Density [g/cm$^3$] | Blaine Surface [cm$^2$/g] |
|---|---|---|
| Cupula slag filler (CSF) | 2.810 | 5112 |

### 3.2.4. Setting Time

Figure 9 shows a graph comparing the initial and final setting time values for the reference cement (CEM III 32.5), for a cement consisting of 80% CEM I + 20% CSF and for another cement consisting of 50% CEM I + 50% CSF.

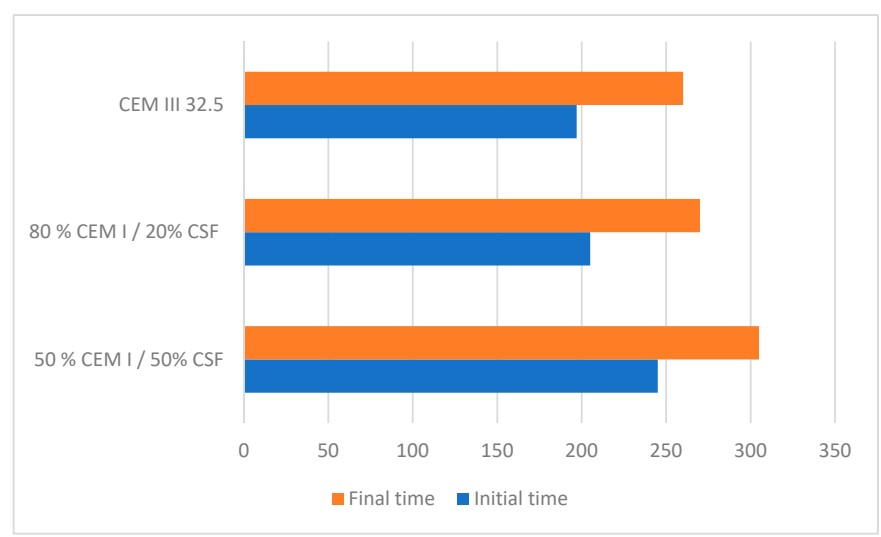

**Figure 9.** Cement setting times without setting retarder.

Based on the results shown in Figure 9, it can be concluded that increasing the percentage replacement of CEM I with CSF delays both the start and end of the setting. To ensure that the cement with 50% CEM I + 50% CSF could be used for silver application, it was decided to use the same type of cement, but with a setting activator (Na$_2$O), see Figure 10.

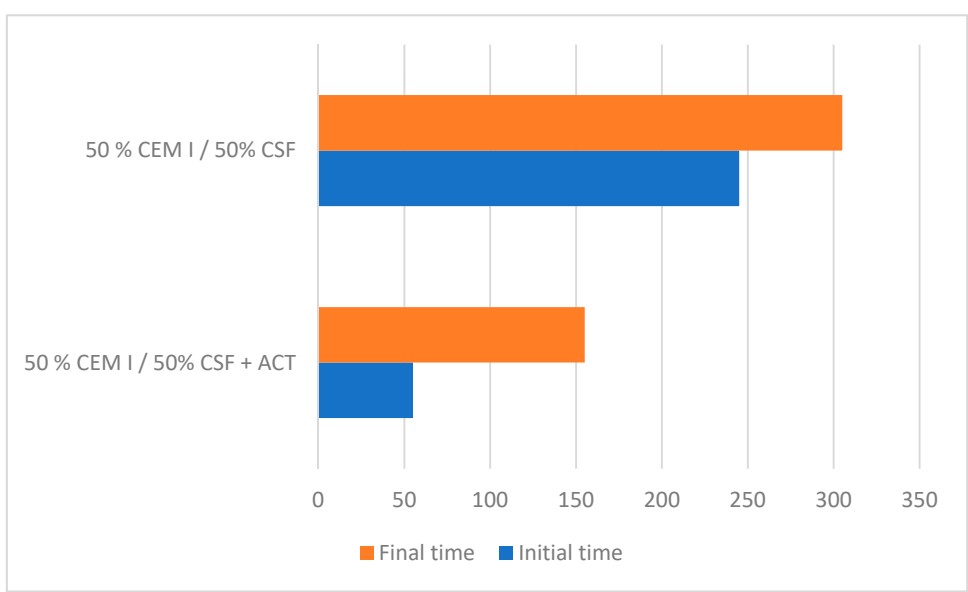

**Figure 10.** Cement setting times with setting retarder.

As can be seen in Figure 10 the increase in the start and end of set time caused by replacing 50% CEM I with CSF can be corrected by using a setting time activator.

### 3.2.5. Vitrification

The pozzolanic reactivity of the slag will become higher as the degree of crystallinity decreases. The high degree of vitrification of the slag used is shown by the lack of crystalline peaks in its X-ray diffractogram, which confirms that it is a highly amorphous material, see Figure 11.

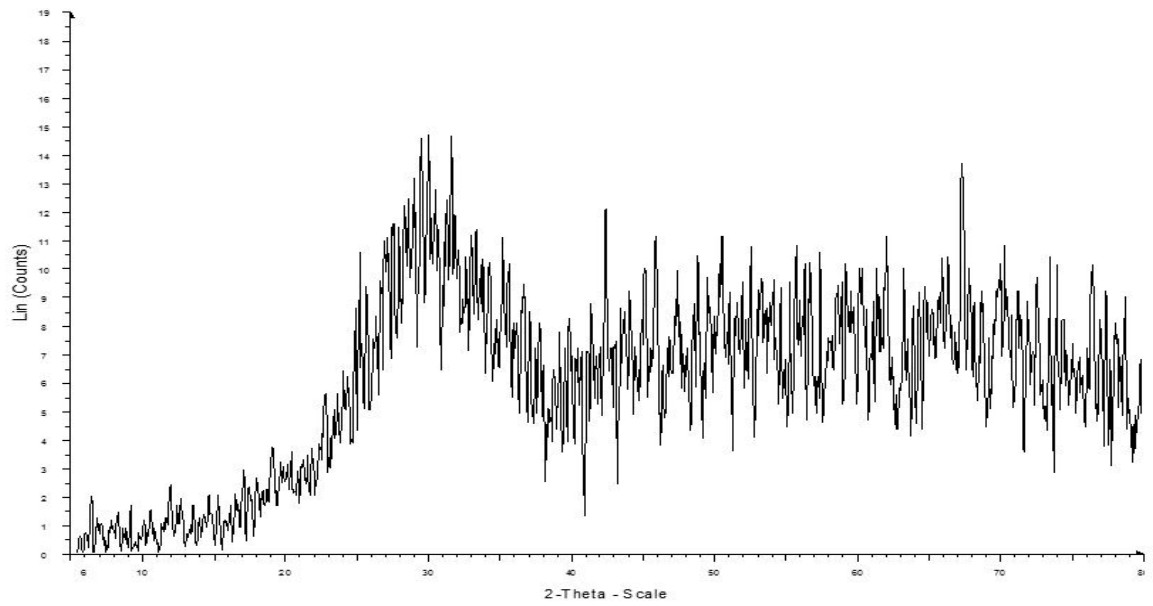

**Figure 11.** X-ray powder diffractogram of the cupola slag.

### 3.2.6. Pozzolanicity Test

Representing the concentrations of hydroxyl ions and calcium hydroxide measured in the solution on the graph provided by standard EN-196-5, the point shown in Figure 12 is obtained. This point represents the saturation concentration in calcium oxide of the solution as a function of the concentration of hydroxyl ions at 40 °C. According to the aforementioned standard, the cement complies with the pozzolanicity test when the point obtained is below the saturation concentration curve in calcium oxide. This is the case of the test carried out, which yielded values of [CaO] = 6.4 and [OH] = 58.0, which allows us to state that the slag analysed complies with the pozzolanicity test.

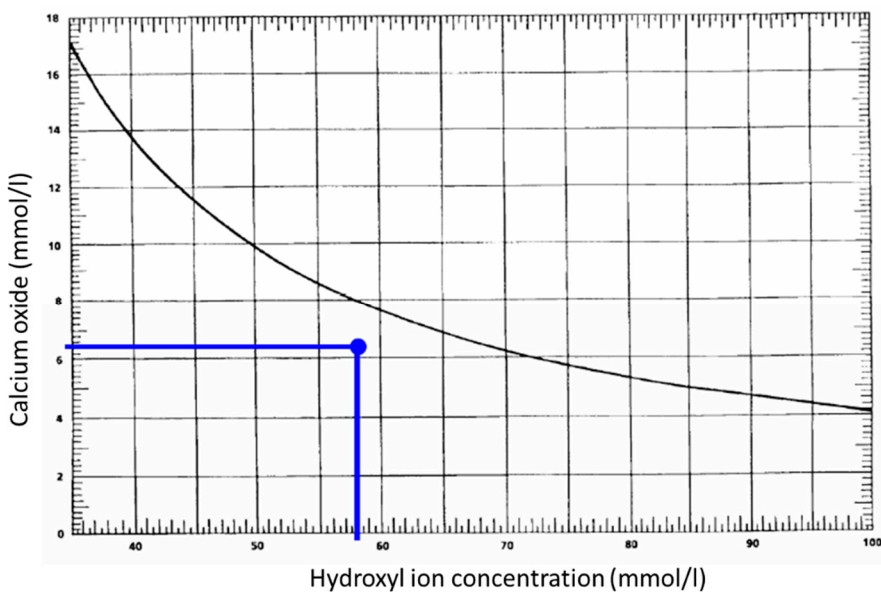

**Figure 12.** Pozzolanicity test results.

### 3.2.7. Resistance to Sulphate

The clinker composition was determined from the Bogue equation, and according to this formulation the $C_3A$ content of the clinker is 7%.

Since EN 197-1 [32] evaluates whether a cement is sulphur resistant on the basis of the cement type (I, III or IV), the first step is to place our new cement in one of these categories. Since the cement type that the CEM I + CSF mixture most closely resembles is type IV, in order to guarantee sulphate resistance, the $C_3A$ must be found to be less than 9%. For this reason, it is considered to be a sulphate-resistant cement, but similarly to CEM IV, the amount of clinker replacement is limited to between 20 and 55%, which is why the replacement percentages analysed in the following sections are 20 and 50% of type I cement.

### 3.3. Mortar Workability

Figure 13 shows the examples of the appearance of the slump table test for the standard mix, for the three mixes corresponding to a 20% replacement of cement by CSF and the four corresponding to a 50% replacement. Figure 14 compares the mean diameter values of the slump table for the different dosages.

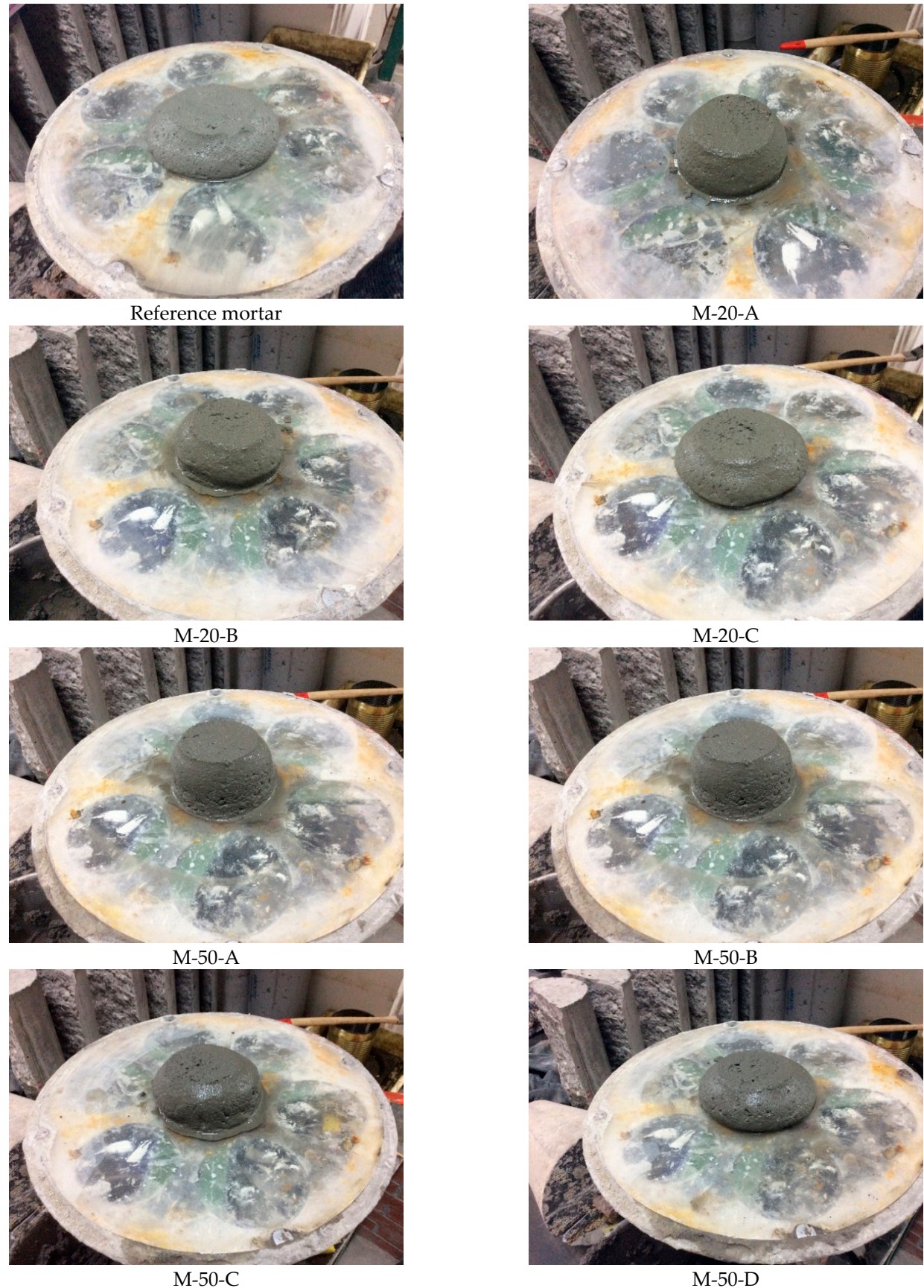

**Figure 13.** Fresh mortar workability.

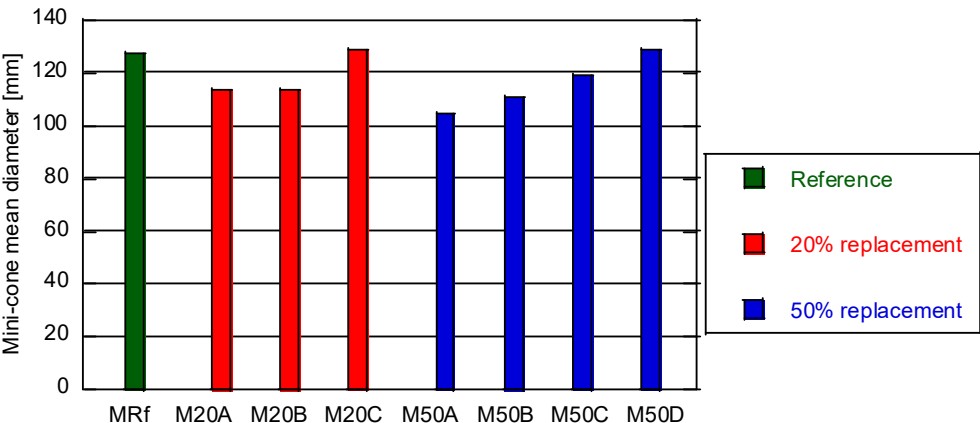

**Figure 14.** Mortar workability, slump table test results.

In order to make the results of the different replacement percentages comparable, it was decided to use mortars with equivalent workability. For this reason, based on the results shown in Figures 13 and 14, M-20-C and M-50-D were selected as comparable mixes to the reference mix.

Based on the results obtained in these tests, it is also possible to estimate the effect of the w/c ratio and the ratio of cement replacement by CSF. To perform this analysis, Figure 15 shows how the slump table diameter varies when modifying the rest of the variables in relation to the reference mix. Figure 15a shows the cases corresponding to 20% replacement. Figure 15b shows the cases corresponding to 50% replacement. Figure 15c shows all cases sorted by replacement percentage. Finally, Figure 15d shows the equations used to obtain each variable represented.

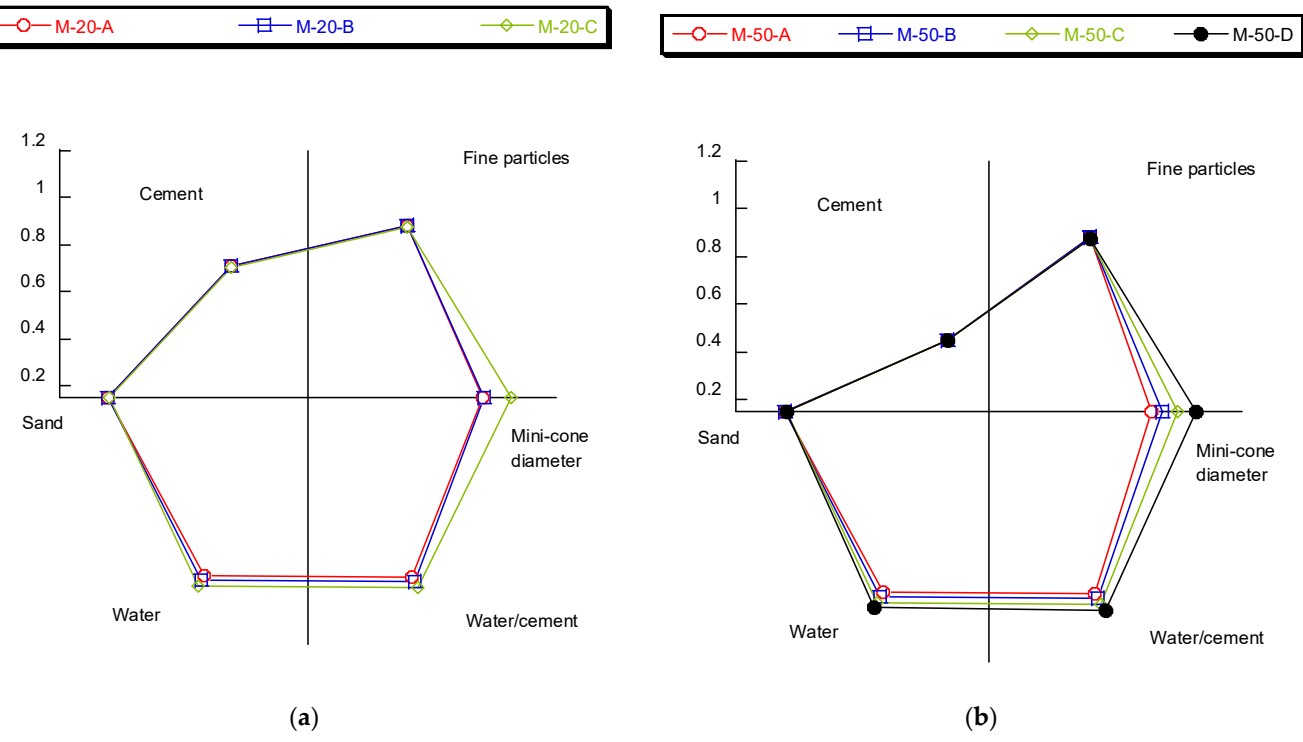

**Figure 15.** *Cont.*

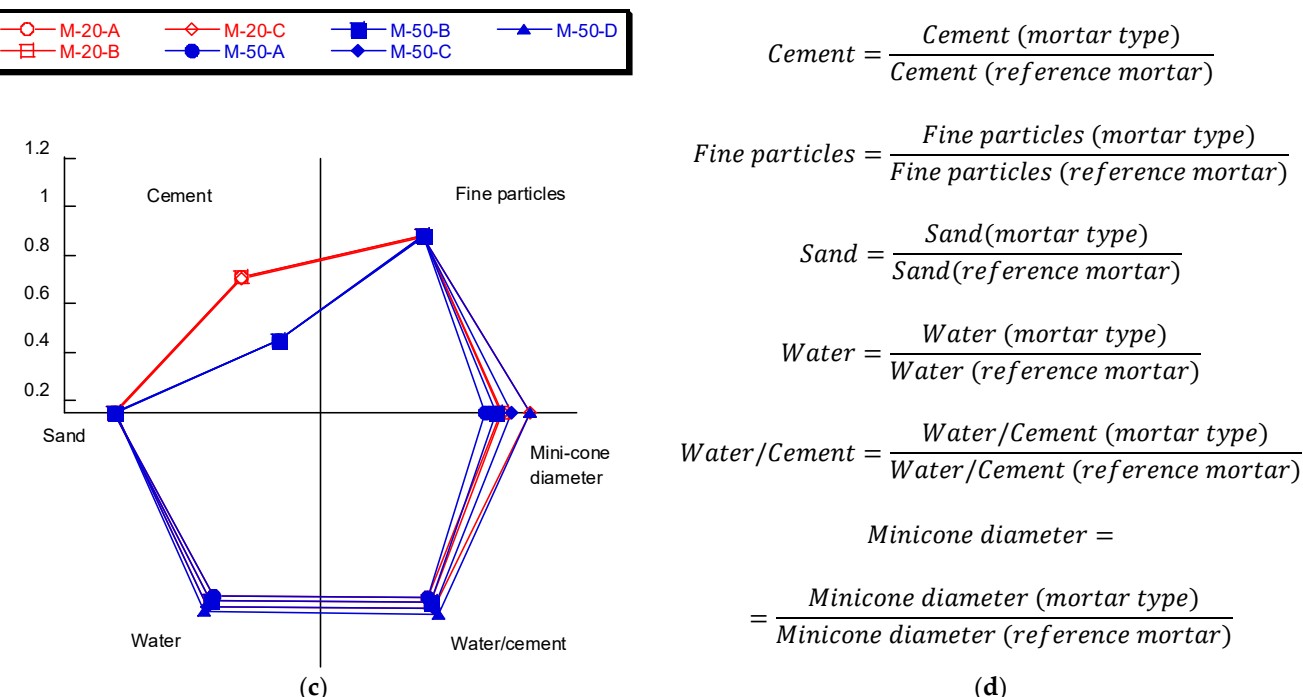

(c)

$$Cement = \frac{Cement\ (mortar\ type)}{Cement\ (reference\ mortar)}$$

$$Fine\ particles = \frac{Fine\ particles\ (mortar\ type)}{Fine\ particles\ (reference\ mortar)}$$

$$Sand = \frac{Sand\ (mortar\ type)}{Sand\ (reference\ mortar)}$$

$$Water = \frac{Water\ (mortar\ type)}{Water\ (reference\ mortar)}$$

$$Water/Cement = \frac{Water/Cement\ (mortar\ type)}{Water/Cement\ (reference\ mortar)}$$

$$Minicone\ diameter =$$

$$= \frac{Minicone\ diameter\ (mortar\ type)}{Minicone\ diameter\ (reference\ mortar)}$$

(d)

**Figure 15.** Influence of the mix proportions on the workability, (**a**) corresponding to 20% replacement, (**b**) 50% replacement, (**c**) all cases sorted by replacement percentage, (**d**) equations used to obtain each variable represented.

The main conclusion drawn from this figure is that increasing the amount of water or the w/c ratio results in a larger diameter in the slump table test. Although this observation is to be expected on the basis of numerous studies, it is possible to appreciate that in the case of 50% replacement, the influence of the w/c ratio is notably greater than in the case of 20% replacement.

### 3.4. Compressive Strength

Figure 16 shows the evolution of the compressive strength as a function of age for the three mortars designed at the ages of 2, 7, 28, 60, and 90 days.

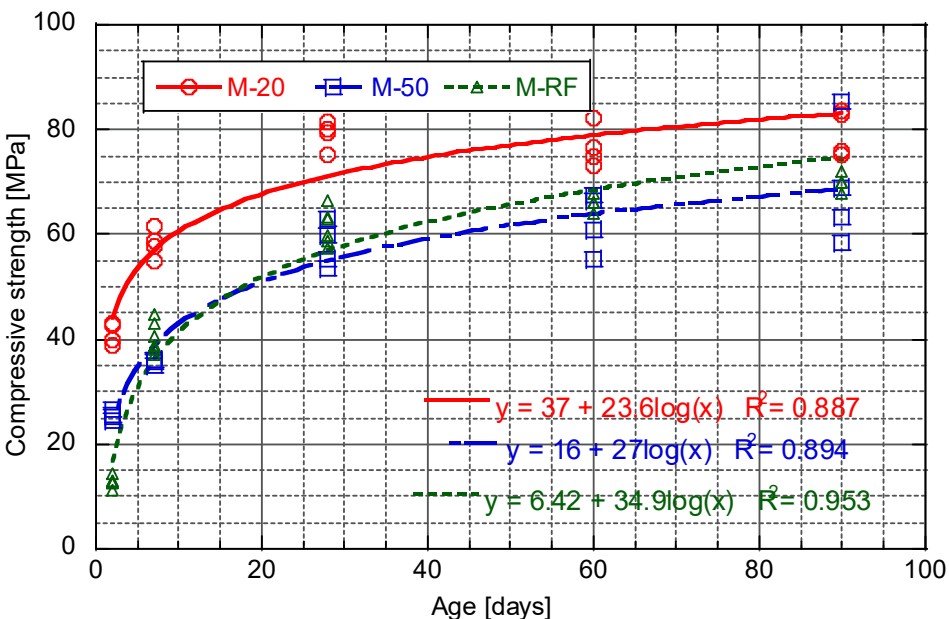

**Figure 16.** Evolution of the compressive strength as function of time.

In Figure 16, it is possible to appreciate that for the age of 2 days, the results corresponding with M-50 are about twice the reference values, while those for M-20 are about three times the reference values. The importance of these values at an early age is that it is possible to handle the tubes without breaking the coating. For this reason, on the basis of the results observed, the replacement of the cement would reduce the waiting times until the pipes can be handled. At the age of 7 days, the reference and M-50 results are almost equal, while the M-20 results are approximately 50% higher than the reference results. For the ages of 28 and 60 days the values for the reference and the M-50 continue to be quite similar, while the difference with the M-20 is reduced. This is because M-20 practically maintains its strength constant while the reference continues to grow slowly. At the age of 90 days the reference mortar has a slightly higher compressive strength than M-50, although it seems that it has already reached a stable value and therefore this difference will not increase. When comparing the results of M-20 with the reference mortar, the M-20 is still higher, approximately 20%.

From analysing Figure 16 it is possible to see how the fit to a logarithmic function is better for early ages, especially in the case of having an M-20 and CEM III. This effect can be justified because the mortar has reached its maximum strength values and does not continue to evolve. This may be because OPC at 28 days has almost completely reacted, which is why the mechanical properties of those mortars with CEM I practically do not evolve beyond 28 days.

It is also possible to appreciate that the early age strengths are better in the case of using a combination of CEM I +CSF than in case of using CEM III. This is of great interest because it is likely that the pipes will need to be handled during the first few days of aging and, due to these better mechanical properties, pipes made with these new cements will be less likely to break. Based on this observation it could be concluded that the increase in the setting time of cement with 50% CEM I + 50% CSF does not have a major impact.

### 3.5. Cement Comparison

Table 7 shows a summary of the main properties characterised for both the different cements designed and the mortars manufactured.

**Table 7.** Comparison of the properties of the different cements and mortars.

| Property | CEM III | 80% CEM I + 20% CSF | 50% CEM I + 50% CSF |
| --- | --- | --- | --- |
| Sulphur resistant | Yes | Yes | Yes |
| Leached problems | No | No | No |
| Setting time | Currently used | Similar to reference | Larger, but compensable |
| Workability | Currently used | Similar to reference | Similar to reference |
| Mechanical properties | Currently used | Higher to reference | Slightly lower than reference |

Neither the reference mix nor the two mixes designed will be affected by the presence of sulphates, a guarantee based on the reduced amount of $C_3A$ provided by the cements. Regarding leachates, a requirement necessitated by a possible application of the mortars for the channelling of drinking water, it has been verified that the raw materials used comply with the requirements defined by the current regulations. Furthermore, in the case of using ground slag to manufacture mortars, as the slag will be encapsulated in the mortar, the risk is even lower. The workability of the mortars was considered as a key parameter for the application studied and, therefore, the *w/c* ratio was set to ensure that the workability of the three mortars was equal.

It was found that the presence of CSF increases both the start and end setting times. It was found that with 80% CEM I + 20% CSF cement the setting time is similar to that of the cement currently used for this application. However, in the case of using a 50% CEM I + 50% CSF cement, the setting time increased significantly, but this increase in the setting time could be corrected by means of a setting activator.

Finally, in relation to the mechanical properties of the three mortars, it can be seen that when the mortars made with the 50% CEM I + 50% CSF cement and the reference cement (CEM III) are compared, they give rise to similar mechanical properties, while the mortars with 80% CEM I + 20% CSF cement have superior mechanical properties.

Based on the above comments, it can be concluded that 80% CEM I + 20% CSF cement would be suitable to replace the currently used CEM III, as in all aspects it provides similar or superior properties. In order to use 50% CEM I + 50% CSF cement, it would be necessary to first ensure: (i) The loss of mechanical properties is acceptable; (ii) to evaluate whether it is necessary to use a setting accelerator.

As a general consideration, the use of these slags as substitutes for cements, and the main motivation for the work, not only represents a notable carbon footprint associated with cement, but also involves the elimination of a waste that is being taken to landfills. From the economic point of view, the company generating the waste will benefit from savings in landfill rates (which have also been increasing in recent years) for thousands of tons per year. The contribution of these slags to the carbon footprint of cement may not be very high since the rates of cement consumption far exceed the volumes of waste generation. This should be good news because it guarantees that all the generated waste could have an application.

## 4. Conclusions

In this work, the substitution of CEM III B 32.5 R cement in the manufacture of mortars for the internal lining of centrifuged cast iron pipes for water channelling was evaluated. The proposal was to reuse the finely ground slag from the manufacturing process as an addition mixed with CEMI 52.5 R cement, which is basically Portland clinker.

The experimental programme showed that the optimum substitution was achieved with a replacement percentage of 20% of the cement and a $w/c = 0.38$. This provided the same workability as the currently used reference mortars, which is crucial because these mortars are also laid by centrifugation. For this reason, it is also important to guarantee a good compressive strength at early ages (2 days), which was more than achieved ($\approx$40 MPa). Furthermore, as the pozzolanicity of the slag was proven, it also contributed to the long-term strength, reaching 80 MPa at 28 days, a value that remained almost unchanged up to 90 days ($\approx$85 MPa).

The nature of the application requires the use of sulphate-resistant cements such as CEM III b. Since the CEM I + 20%CSF mixture resembles a CEM IV, it was verified by means of Bogue's formulation that the amount of CA3 is less than 9% in the clinker of CEM I, which is the limit set by current regulations for CEM IV to be sulphate resistant. This guarantees that the CEM I + 20%CSF mixture is also sulphur-resistant.

The use of this type of cement for the internal coating of centrifugally spun cast iron pipes for water channelling makes it mandatory to control the potability of the water flowing through the pipes. To ensure that the valorisation of the cupola slag does not contaminate the water, leachate tests were carried out and, in all cases, they were found to be within the current regulations.

The reuse of slag from a metallurgical process in this application has a double advantage: (1) environmental, as it prevents this waste going to landfill, as has happened up to now, and (2) economic, as it saves on transport and landfill fees and at the same time reduces costs by saving on cement (less quantity and use of cheaper cement). With a moderate investment in a plant for slag crushing, easily automated, a convergent application to the much desired circular economy of the process is achieved, which makes it much more sustainable.

**Author Contributions:** Conceptualization, C.T., J.S., J.A.P.; methodology, C.T., J.S.-A., I.S., J.S., J.A.P., A.C.; validation, C.T., J.S., J.A.P.; formal analysis, C.T., J.S.-A., I.S., J.S., J.A.P., A.C.; investigation, C.T., J.S.-A., I.S., A.C.; resources, C.T., J.A.P.; data curation, J.S.-A., I.S., A.C.; writing—original draft preparation, C.T., J.S.-A.; writing—review and editing, C.T., J.S.-A., I.S., J.S., J.A.P., A.C.; supervision,

C.T., J.S., J.A.P., A.C.; project administration, J.S., J.A.P. All authors have read and agreed to the published version of the manuscript.

**Funding:** This research received no external funding.

**Institutional Review Board Statement:** Not applicable.

**Informed Consent Statement:** Not applicable.

**Data Availability Statement:** Not applicable.

**Acknowledgments:** The authors would like to thank: LADICIM, the Laboratory of Materials Science and Engineering of the University of Cantabria, for making the facilities used in this research available to the authors. The authors would like to acknowledge the "Augusto Gonzalez Linares" postdoctoral grant programme of the University of Cantabria for support.

**Conflicts of Interest:** The authors declare no conflict of interest. The funders had no role in the design of the study; in the collection, analyses, or interpretation of data; in the writing of the manuscript, or in the decision to publish the results.

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
