# Peer review of "Physical-Mechanical Properties of Cupola Slag Cement Paste"

_applsci, doi:10.3390/app11157029_

Round 1
Reviewer 1 Report
The search for new ingredients for the production of concrete is very important from the technological and environmental point of view. The use of cupola slag is interesting in terms of replacing Portland cement. Below are some comments and suggestions:
- In the Introduction, please mention that slags can also be used in mining technologies such as backfilling, which is desirable for environmental reasons (doi.org/10.3390/en14113186).
- In the Chapter 2.1.2, it should be stated what was the result of the 8-hour period (lines 128 and 130).
- The word "Equation 1" at the end of the equation is not necessary.
- Figure 2, although it says in line 163, the composition curve was determined according to the EN 933-1 standard, but it would be useful to have two or three sentences describing this method in the tests.
- Chapters 2.2.6; 2.2.7; 2.2.8 contain only two sentences, they should be expanded or merged with another chapter.
- Chapter 2.2.7 it should be written what the limiting dimensions in this test are.
- In the Chapter 2.2.8 it is necessary to write what dimensions the samples had and what was the load rate.
- Table 2, it should be written how Blaine Surface was specified.
- Figure 7a and Table 4 show the same results.
- Figure 8 should contain descriptions of the horizontal and vertical axes.
- Chapter 3.2.5 contains only two sentences, it should be expanded by a few sentences.
- Figure 15, descriptions should be attached to the axis.
Author Response
Manuscript ID: applsci-1309698
Authors’ replies to reviewers
Dear Editor,
We wish to thank you for the opportunity to improve our paper ‘Physical-mechanical properties of cupola slag cement paste’ further to the reviewers’ suggestions. Our replies, specifying the amendments introduced in the revised MS, are set out below. All the changes are highlighted in the new version for readier identification.
Best regards,
Carlos Thomas
(for all co-authors)
Authors’ replies to reviewers’ comments
Reviewer 1
- In the Introduction, please mention that slags can also be used in mining technologies such as backfilling, which is desirable for environmental reasons (doi.org/10.3390/en14113186).
Thank you very much for your comment, the following text was added including the requested reference lines 49 to 50: “This waste is also used in mining technologies such as backfilling, which is desirable for environmental reasons [6]”.
- In the Chapter 2.1.2, it should be stated what was the result of the 8-hour period (lines 128 and 130).
Thank you very much for your comment, the following text was added, lines 140-143 “It was decided to crush 8 h after testing different times and obtaining the minimum time required to reduce the material size to filler. 2h, 4h, 6h and 8h were tested and it was found that it is necessary 8h, in an enamel mill with alumina balls, to obtain more than the 95%wt. of the material smaller than 75 mm.”
- The word "Equation 1" at the end of the equation is not necessary.
Thank you very much for your comment. The indicated modification was made.
- Figure 2, although it says in line 163, the composition curve was determined according to the EN 933-1 standard, but it would be useful to have two or three sentences describing this method in the tests.
Thank you very much for the comment, the following text was added: "The procedure starts by selecting a sample of material, which is introduced into the oven until it reaches a constant weight. Once the sample is completely dry, it is sieved and each of the fractions obtained is weighed".
- Chapters 2.2.6; 2.2.7; 2.2.8 contain only two sentences, they should be expanded or merged with another chapter.
The three sections were merged into one: "Mortar manufacture, curing conditions and testing procedure".
- Chapter 2.2.7 it should be written what the limiting dimensions in this test are.
Thank you very much for the comment, the following text was added: “For the determination of the consistency of the mortars, the procedure according to EN 1015-3 - Methods of test for mortar for masonry - Part 3: Determination of consistency of fresh mortar (by flow table) [40] is followed. Very similar to that used to determine the consistency of concrete by means of a shaking table. In the case of concrete [41], there is a limitation regarding the maximum size of the aggregate of 64 mm. In the case of mortar, the limitation is 4 mm. In both tests, the fresh cone is shaken 15 times to finally measure the diameter in two perpendicular directions.”
- In the Chapter 2.2.8 it is necessary to write what dimensions the samples had and what was the load rate.
In the previous section 2.2.8. it was already stated that these were standardised tests, i.e. prismatic specimens of 160 x 40 x 40 mm and at a loading ratio of 50 Nm/s. In any case, to make it easier for future readers this information was added on the text: “The compressive strength was determined according to EN 196-1 [39] (prismatic specimens of 160 x 40 x 40 mm and at a loading rate of 50 Nm/s).”.
- Table 2, it should be written how Blaine Surface was specified.
This information was added in the chapter 2.1.1. Lines 108-111 “The density and Blaine specific surface area were determined according to UNE 80103 [33] and EN 196-6 [34] respectively.”
- Figure 7a and Table 4 show the same results.
These are indeed duplicate results, so figure 7a was deleted.
- Figure 8 should contain descriptions of the horizontal and vertical axes.
You are right, it may be an unintuitive figure to analyse. In this figure the Y axes have no dimensions because the value they indicate is each of the elements (what appears in the legend). This is in order to be able to plot all the data in a single figure. Regarding the X axes, due to the large amount of information present in this figure, when drawing the values, they looked very small, so it was decided to put in the lower part of the figure the range of months to which it corresponds on a linear scale. If the reviewer thinks it is correct, we, the auotres, prefer to keep the image as it is, but adding the following explanation: “In this figure, the Y-axes indicate the quantity of each component, see legend. For the X-axis, this is a linear scale of the time indicated at the bottom of the figure.”.
- Chapter 3.2.5 contains only two sentences, it should be expanded by a few sentences.
All the short sections have been merged together with one another.
- Figure 15, descriptions should be attached to the axis.
In radar plots all axes have the same scale. This is because in this type of plots the evolution of each of the variables can be seen from a reference value (it is a ratio between the corresponding value and the reference value). Therefore, all values are in the same range of values, close to 1, and are dimensionless. This information is defined in Figure 15 (d).

Reviewer 2 Report
Dear Authors,
just a few comments. Can you evidence more the sustainable advantages, in terms of cost reduction for the cost of cement and for example the green impact? this is important to evidence and make in place practically the research goals. thanks
Author Response
Manuscript ID: applsci-1309698
Authors’ replies to reviewers
Dear Editor,
We wish to thank you for the opportunity to improve our paper ‘Physical-mechanical properties of cupola slag cement paste’ further to the reviewers’ suggestions. Our replies, specifying the amendments introduced in the revised MS, are set out below. All the changes are highlighted in the new version for readier identification.
Best regards,
Carlos Thomas
(for all co-authors)
Authors’ replies to reviewers’ comments
Reviewer 2
- just a few comments. Can you evidence more the sustainable advantages, in terms of cost reduction for the cost of cement and for example the green impact? this is important to evidence and make in place practically the research goals. Thanks.
Thank you very much for your suggestion, we agree with the reviewer and the following paragraph has been included: As a general consideration, the use of these slags as substitutes for cements, and the main motivation for the work, not only represents a notable carbon footprint associated with cement, but also involves the elimination of a waste that is being taken to landfills. From the economic point of view, the company generating the waste will benefit from savings in landfill rates (which have also been increasing in recent years) for thousands of tons per year. The contribution of these slags to the carbon footprint of cement may not be very high since the rates of cement consumption far exceed the volumes of waste generation. This should be good news because it guarantees that all the generated waste could have an application.
